

**Brief communication: Use of lightweight and low-cost steel net**
**electrodes for electrical resistivity tomography (ERT) surveys**
**performed on coarse-blocky surface environments**
Mirko Pavoni[1], Luca Peruzzo[1], Jacopo Boaga[1], Alberto Carrera[1], Ilaria Barone[1] and Alexander Bast[2,3]
[1] Department of Geosciences, Università degli Studi di Padova, Padova, Italy.
[2] WSL Institute for Snow and Avalanche Research SLF, Permafrost Research Group, Davos Dorf, Switzerland.
[3] Climate Change, Extremes and Natural Hazards in Alpine Regions Research Center CERC, Davos Dorf, Switzerland.
*Correspondence to*: Mirko Pavoni (mirko.pavoni@unipd.it)
**Abstract.** ERT is a widely used geophysical technique for characterizing various mountainous environments where land
surfaces consist of coarse blocks and debris, such as landslide deposits or rock glaciers. In this situation, installing the common
steel spike electrodes is both challenging and time-consuming, and achieving galvanic contact between the electrodes and the
surface is difficult. In this work, we have successfully tested an alternative electrode that is tougher, lighter and cheaper than
the recently proposed conductive textile electrode. A thin stainless-steel net and sponges are used to create small bags that can
be easily inserted between the blocks, and then removed.
**1 Introduction**
Electrical Resistivity Tomography (ERT) is one of the most widely used geophysical methods for the characterization of
different types of study environments (e.g. Boaga et al., 2018; Deiana et al., 2022; Carrera et al., 2024; Pavoni et al., 2024;
Peruzzo et al., 2024; Uhlemann et al., 2024) as data acquisition is usually relatively rapid and not particularly complicated
(Binley, 2015). Thanks to the development of open-source, robust and reliable inverse modeling techniques, the distribution
of electrical properties in the investigated subsurface can be easily reconstructed (Rücker et al., 2017). The reliability of the
obtained resistivity models is linked to the quality of the acquired data, and one of the key aspects of optimizing the acquisition
of ERT measurements is to guarantee a good galvanic coupling between electrodes and soil (Pavoni et al., 2022). The parameter
that quantifies the quality of the galvanic coupling is defined as contact resistance, i.e. the resistance to electric current flow
across the contact surface between the electrode and the ground (Binley and Slater, 2020). For this reason, electrodes must be
made from a material with high electrical conductivity. The metals commonly used include graphite, copper, and stainless
steel (Rücker and Günther, 2011). Graphite guarantees a very low resistance to the flow of electric current but has very poor
mechanical properties. Copper is the best electrical conductor among the specified metals; however, it tends to oxidize and
consequently loses its conductive properties quickly over time. On the other hand, stainless steel has high electrical
conductivity, albeit lower than graphite and copper, which does not diminish over time and guarantees excellent mechanical
properties. Furthermore, stainless steel is less expensive compared to copper and graphite (Reynolds, 2011). In light of this,
for decades, electrodes for ERT surveys have typically been made as round stainless-steel spikes measuring 30-40 cm in length
and 1-2 cm in diameter (Rücker and Günther, 2011). Hammering these electrodes into fine soils and sediments is
straightforward, but their installation becomes challenging and time-consuming in coarse-blocky environments, such as
landslide deposits or rock glaciers (Buckel et al., 2023). Furthermore, their removal can also be complicated if the electrodes
are firmly embedded between the blocks. Lastly, although good physical coupling between the electrodes and the coarse-
blocky surface is guaranteed, sponges soaked in salt water are added to improve contact resistance (Pavoni et al., 2022).
To facilitate and expedite the installation of the electrode array on rock glaciers, Buckel et al. (2023) proposed a type of
electrode made from conductive textile and filled with sand. The fist-sized textile sachets can be easily placed between the



coarse blocks or boulders, wetted with salt water and finally easily removed. The reliability of these textile electrodes was
tested by Bast et al. (2024). Nevertheless, the conductive textile used in both studies primarily comprises copper and nickel as
conductive materials, which are two metals that are susceptible to oxidation. Therefore, the textile may lose its conductive
properties after a while (Bast et al. 2024). Replacing deteriorated electrodes involves considerable expense, as the cost of a
conductive textile electrode is roughly € 15 each (Buckel et al. 2023). Further, the weight of the textile electrodes is comparable
to the one of the traditional stainless-steel spikes (~ 250-300 g, Bast et al. 2024). Given these limitations, we propose a
lightweight stainless-steel net electrode that is readily available at low cost and has a carwash sponge inside the formed sachet
instead of sand. The used material ensures that the electrode is not prone to oxidation issues, significantly reduces its weight,
and greatly enhances the mechanical strength of the sachets compared to textile electrodes.
In this work, we successfully verified the reliability of the newly proposed stainless-steel net electrode by performing ERT
measurements at the same test sites with coarse-blocky surfaces as used in Bast et al. (2024). We compared the performance
and results with those obtained from parallel measurements with traditional stainless-steel spikes.

## 51  2 Site description

The proposed stainless steel-net electrodes were tested in typical (high) mountain environments characterised by surfaces
composed of coarse blocks and debris: a landslide deposit (Marocche di Drò), an inactive rock glacier (Sadole rock glacier),
and an active rock glacier (Flüela rock glacier). For details, maps, and images of the study sites, we refer to Bast et al. 2024.
The landslide deposit known as Marocche di Drò (Trento, Italy, 45.983 N, 10.941 E) is composed of a chaotic arrangement of
calcareous blocks and debris (limestone) on the surface and a more heterogeneous sediment body beneath (Weidinger et al.,
2014). The investigation line of the comparison test is identical to the one used by Bast et al. (2024).
The Sadole rock glacier (Trento, Italy, 46.242 N, 11.592 E) has a surface with large blocks and coarse debris of ignimbritic
volcanic rocks (Pavoni et al. 2023). Bast et al. (2024) confirmed that the central lobe contains a frozen layer at about 10 m
depth, and our comparative electrode test was performed solely on the first half of the former investigation line.
The Flüela rock glacier (Grisons, Switzerland, 46.746 N, 9.951 E) presents a surface composed of a chaotic mixture of
metamorphic blocks and boulders (amphibolites and paragneisses) and isolated areas with finer sediment (Boaga et al., 2024).
The comparative electrode test was performed in the upper half of the investigation line used by Bast et al. (2024).

## 64  3 Methods

### 65  3.1 Stainless steel-net electrodes

As shown in Fig 1a, the proposed steel-net electrode is created by cutting square sheets measuring 35 x 35cm from a thin
commercial stainless-steel net, which is moulded into a fist-sized bag containing a car-wash sponge. Finally, the bag is sealed
with a thin electrician's cable tie. Each electrode weighs about 50 g (about one-fifth of steel-spike and textile electrodes), and
the material costs about € 3-4 euros (approximately one-fourth compared to the steel-spike and textile electrodes).

### 70  3.2 Data acquisition

Data were collected with a Syscal-Pro georesistivimeter (Iris Instruments, Orléans, France, www.iris-instruments.com), with
24-electrode arrays, and using different spacings at each test site: 5 m at Marocche di Drò, 3.5 m at Sadole, and 2 m at Flüela.
The acquisition scheme used was a dipole-dipole configuration with different skips (Pavoni et al., 2023), and included
reciprocal measurements (a total of 522 measured quadrupoles). For each of the three investigation lines, measurements were
acquired using the traditional stainless-steel spike electrodes (coupled with sponges - Fig. 1b), and subsequently with the
proposed stainless steel-net electrodes (Fig. 1c). In both cases, we used salt-water to improve the galvanic contact between the
electrodes and the coarse blocky surface (Pavoni et al., 2022). The two different electrode types were positioned at the same



locations between the blocks and/or boulders (Fig. 1b-1c), and the same volume of salt water was used to wet them (~ 0.5 l for
each electrode before each acquisition).

## 3.3 Data processing

At each test site, we measured the contact resistances before acquiring the datasets with the different types of electrodes. To
compare the contact resistances, we plotted the values obtained for each pair of electrodes using a histogram. To evaluate the
quality of the acquired datasets, the reciprocal error (Tso et al., 2017) of each quadrupole was calculated, and histograms were
used to compare the values obtained from the different electrode types. The reciprocal error values were grouped in equal
intervals of 1 % from 0 – 10 %. Values exceeding 10 % were grouped into one bin. The apparent resistivity values were
compared using a scatterplot, plotting a simple regression line and including the $R^2$-value.
At each test site, the datasets were filtered using a reciprocal error threshold that allowed to obtain a section of apparent
resistivities with a homogeneous distribution of measurement points (Pavoni et al., 2023). After filtering, the inversion process
was performed with the open-source software ResIPy (Blanchy et al., 2020). The results obtained from the different electrodes
were compared by both, plotting the inverted resistivity models and the resistivity values obtained in each cell of the inversion
mesh on a scatterplot, including a simple regression line and the $R^2$-value.

## 4 Results and data interpretation

In the comparative test of the Marocche di Drò, the contact resistances measured with the two types of electrodes were very
similar (Fig. 2a), with values < 200 kΩ (most being < 100 kΩ), ensuring optimal conditions to acquire ERT measurements in
this challenging environment (Pavoni et al., 2022). This is confirmed by checking the quality of the acquired data (Fig. 2b).
Most of the measured quadrupoles (85 % for spike and 90 % for net electrodes) show a reciprocal error < 5 %, which was
chosen as a threshold to filter the datasets and used as the expected error of the datasets in the inversion processes. As
highlighted in the scatterplots (Fig. 3a and 3d), the measured apparent resistivities and the inverted resistivities (Fig. 3g and
3j) show a very high correlation (apparent resistivity $R^2 = 0.99$, inverted resistivities $R^2 = 0.99$). The obtained inverted
resistivity sections are nearly identical and allow reconstructing the known structure of the landslide deposit (Weidinger et al.,
2014), where large blocks with extensive air voids are characterized by high resistivity values and located near the surface. At
greater depths, the resistivities drop drastically, confirming the presence of more heterogeneous and finer sediments.
At the Sadole rock glacier, the measured contact resistances were significantly improved using the net-electrodes (Fig. 2c).
With the spike-electrodes array, almost half of the contact resistances have values > 100 kΩ, and > 50 % of the values are
higher > 200 kΩ. In contrast, in the case of net electrodes, the values are clearly lower and allow the acquisition of a higher-
quality dataset (Fig. 2d). In the dataset acquired with the net electrodes, more than half of the quadrupoles have a reciprocal
error < 1 % and only four quadrupoles have a reciprocal error > 5 %. Nevertheless, the spike-electrodes also allow us to
measure a good-quality dataset, with 90 % of the quadrupoles having a reciprocal error < 5 %. Despite the minor differences
in data quality, the same threshold was applied to both datasets by filtering the quadrupoles with reciprocal error > 5 %, which
was also used as the expected error of the datasets in the inversion processes. As shown in Figures 3b and 3e, the measured
apparent resistivities and the inverted resistivities obtained in the models show an excellent correlation, although slightly lower
than in the Marocche di Drò site. In the first case $R^2 = 0.91$, while for the inverted resistivities $R^2 = 0.93$. Again, the inverted
resistivity sections (Fig. 3h and 3k) are nearly identical and confirm the presence of a frozen layer with high resistivity values
at about 10 m depth (Bast et al., 2024).
The measured contact resistances at the Flüela rock glacier site (Fig. 2e) were generally higher than at the other two sites. With
both electrode types, more than half of the contact resistances have > 200 kΩ. Consequently, the quality of the acquired datasets
is lower (Fig. 2f). In both datasets, only 78 % of the acquired quadrupoles have a reciprocal error < 5 %, and 90 % have an
error < 10 %, which was chosen as the expected error for both datasets in the inversion processes. The correlation between the



measured apparent resistivities with different electrode types is lower than at the other sites (Fig. 3c, $R^2 = 0.8$). Consequently,
the inverted resistivities calculated in the models also show a lower correlation ($R^2 = 0.88$; Fig. 3f). The inverted resistivity
sections (Fig. 3i and 3l) obtained with the different electrode types highlight the same subsurface structure featuring a high
resistivity permafrost body at a few meters depth in the first half of the array (x < 20 m). However, the tomograms reveal slight
differences at the bottom of the models, where the high resistive permafrost body in Figure 3l shows higher values compared
to Figure 3i, and lower values towards the front of the array (x > 25 m) in the unfrozen area.

## 4 Discussion and conclusions

From the results, it is clear that, as is notoriously known, the lower the contact resistances, the higher the data quality. Among
the different test sites, the contact resistances improved the most at the Marocche di Drò, and the data quality was the highest.
At Sadole rock glacier, the contact resistance values were also good, especially those obtained with the net-electrodes, and the
resulting data quality remains very high. On the other hand, at the Flüela rock glacier, the contact resistances were slightly
higher (with both types of electrodes). Consequently, the quality of the acquired data was lower than at the other two test sites.
The findings for the tested conductive textile electrodes versus spike electrodes are similar (Bast et al., 2024): contact resistance
was significantly lower with textile electrodes at the Marocche di Drò and Sadole sites but showed no statistically significant
difference at the Flüela site. Despite this, at the Flüela rock glacier, it was possible to filter the datasets with a reciprocal error
of 10 %, which can be considered an acceptable value in this challenging environment. Nevertheless, the slight differences
observed among the sites can be attributed to the varying lithologies at each site, as limestone (Marocche die Drò) generally
offers higher galvanic contact than ignimbrite (Sadole) and amphibolite or paragneiss (Flüela) (Duba et al., 1978).
In addition to the strong relation between contact resistances and data quality, it is evident that the better the contact resistances
and data quality, the greater the correlation between the measured apparent resistivities with the two types of electrodes and,
consequently, between the calculated inverted resistivities. At the Marocche di Drò, where we obtained the best contact
resistances and data quality, the apparent resistivities measured in the two datasets are nearly identical, as are the inverted
resistivity values in the models. At the Sadole rock glacier, where the contact resistances are slightly higher and the data quality
slightly lower, the correlation between the measured apparent resistivities and the inverted resistivities of the models, although
excellent, is slightly lower than at the Marocche di Drò. As for Marocche di Drò site, the inverted resistivity models obtained
with the two different electrode types are nearly identical at the Sadole site as well. Finally, at the Flüela rock glacier, where
the contact resistances are clearly higher, and the data quality is poorer, the correlation between the measured apparent
resistivities decreases and, consequently, also that one between the inverted resistivities in the models. Nonetheless, the $R^2$-
value (0.88) of the inverted resistivities is quite high and, from the two resistivity models, it is possible to delineate the same
high-resistivity structure of the permafrost in the near subsurface.
In conclusion, the proposed stainless steel-net electrodes yield results equivalent to those achieved with conventional steel-
spike electrodes and, consequently, as demonstrated recently, also with conductive textile electrodes (Buckel et al. 2023;
Bast et al. 2024). The steel-net electrodes maintain all the advantages of the textile electrodes, i.e., facilitating and speeding
up the preparation of ERT investigation lines in environments with blocky surfaces without compromising the quality of the
final result, but, at the same time, they overcome the disadvantages of the conductive textile. The stainless-steel net is more
durable (no oxidation), significantly cheaper and lighter (easier to carry in challenging mountain environments), and is
mechanically tougher than the conductive textile, which can tear more easily when pushed between the blocks.
Future development of this work is to test the stainless-steel net electrodes for induction polarisation measurements, both in
the time and frequency domain.



**Figure 1: a) The stainless steel-net electrodes are made by cutting square sheets (35 x 35 cm) from a thin commercial stainless-steel net, placing a car-wash sponge inside, and closing the bag with an electrician's cable tie. b) A common stainless steel-spike electrode with a sponge saturated with salt water from the array used at the Marocche di Drò site. c) The proposed stainless steel-net electrode wetted with salt water from the array used at the Marocche di Drò site.**

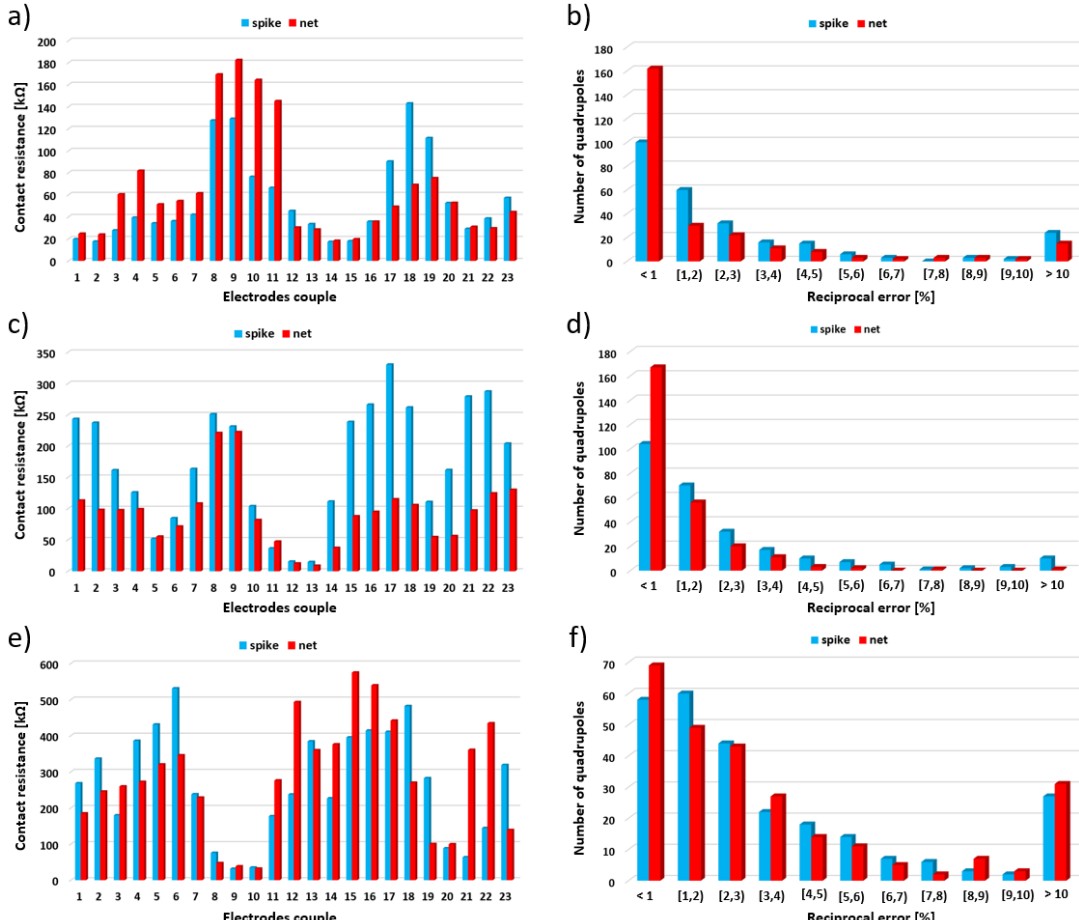

**Figure 2: The histograms (a), (c), and (e) illustrate the comparison of contact resistances [kΩ] recorded respectively at the Marocche di Drò, Sadole, and Flüela test sites with the traditional spike electrodes made of stainless steel and combined with sponges (blue bins) and the newly proposed stainless-steel net electrodes that include a sponge (red bins). The histograms (b), (d) and (f) display the comparison of the reciprocal error [%] of the quadrupoles measured respectively at the Marocche di Drò, Sadole, and Flüela test sites with the traditional spike electrodes made of stainless steel and combined with sponges (blue bins) and the newly proposed stainless-steel net electrodes that include a sponge (red bins). Both types of electrodes were wetted with the same amount of salt water and approximately placed at the same locations between the blocks and boulders (Fig. 1b and 1c).**



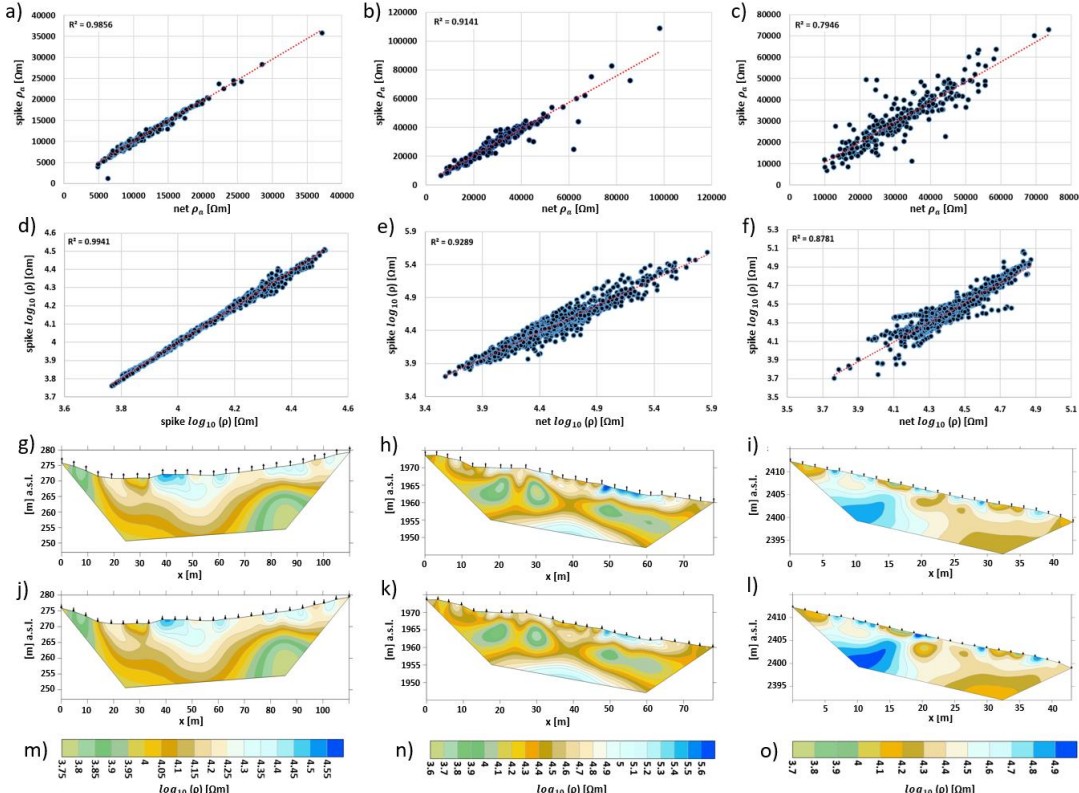

**Figure 3: a) Scatterplot with corresponding regression lines (red dotted lines) and R²-values of the apparent resistivity values (ρa) measured at Marocche di Drò test site with the traditional spike electrodes composed of stainless-steel (coupled with sponges) and the newly proposed stainless-steel net electrodes. d) Scatterplot with corresponding regression lines (red dotted lines) and R²-values of the inverted resistivity values (ρ) obtained from the datasets measured at Marocche di Drò test site with the traditional stainless steel-spike electrodes (coupled with sponges) and the proposed stainless steel-net electrodes. g) Inverted resistivity model obtained from the datasets measured with the traditional stainless steel-spike electrodes at the Marocche di Drò test site. j) Inverted resistivity model obtained from the datasets measured with the proposed stainless steel-net electrodes at the Marocche di Drò test site. m) Color bar scale for the inverted resistivity models of the Marocche di Drò test site. b), e), h), k), and n) are as a), d), g), j), and m) but for the Sadole rock glacier test site. c), f), i), l), and o) are as a), d), g), j), and m) but for the Flüela rock glacier test site.**

*Data Availability Statement.* The datasets used to obtain the results presented in this work are available at the open source repository https://zenodo.org/records/14651003. Furthermore, the ERT datasets will also be included in the International Database of Geoelectrical Surveys on Permafrost (IDGSP).

*Author contributing.* MP initiated and conceptualised the study concept and performed the data processing. All authors were involved in the data acquisition and contributed to the writing and editing of the manuscript.

*Competing interests:* The contact author has declared that none of the authors has any competing interests.

*Financial support:* This study was carried out within the project of the excellence program: "The Geosciences for Sustainable Development" project (Budget Ministero dell'Università e della Ricerca–Dipartimenti di Eccellenza 2023–2027 C93C23002690001), and within the project PRIN 2022 "SUBSURFACE – Ecohydrological and environmental significance of subsurface ice in alpine catchments" (code no. 2022AL7WKC, CUP: C53D23002020006), which received funding from the European Union NRRP (Mission 4, Component 2, Investment 1.1 – D. D. 104 2/2/2022).



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
