# Peer review of "Brief communication: Use of lightweight and low-cost steel net"

_EGUsphere, 2025_

## Referee Comment (RC1)

Reviewer Recommendation and Comments for Manuscript egusphere-2025-405

**Brief communication: Use of lightweight and low-cost steel net electrodes for electrical resistivity tomography (ERT) surveys performed on coarse-blocky surface environments**

**Summary**

The manuscript presents a study on a newly developed electrode design, i.e., stainless steel-net electrodes, which the authors propose for electrical resistivity tomography (ERT) measurements in coarse blocky environments. The electrodes facilitate and accelerate ERT surveys and are cheaper and lighter compared to conventional stainless-steel spike electrodes. The authors demonstrate that contact resistances and reciprocal errors are lower when using stainless-steel net electrodes compared to conventional stainless-steel spike electrodes. Correlations of apparent resistivity values between the two electrode types are high for an exemplary landslide deposit and an Italian rock glacier (with an $R^2$ between 0.91 and 0.99) and slightly lower for a Swiss rock glacier ($R^2$=0.8). The inversion results are similar for both electrode types and successfully reconstruct the known internal structure of the landforms. The results are clearly presented and highlight the relevance of lightweight, easily deployable equipment in harsh alpine terrain with limited accessibility.

However, upon closer examination, the manuscript offers limited novelty and primarily reiterates concepts and methodologies previously presented, particularly in the study by Bast et al. (2024). One of the main concerns lies in the similarity between the figures of the two articles. The representations and analysis of the data are almost identical, with no substantial additions, enhancements or new interpretations. For instance, the authors could have included additional pseudosections to visualize the spatial consistency in the apparent resistivity readings as well as the position of the removed quadrupoles for the different electrode types. In my opinion, the results related to the application of stainless-steel net and textile electrodes could have been presented together in one publication, as the study sites are identical and the data analysis and structure is very similar.

To strengthen the manuscript, I recommend that the authors explore the usability of such stainless-steel net electrodes for induced polarization measurements, as suggested in the Discussion section. Such an investigation would offer a clear advancement over previous research. Another benefit highlighted by the authors consists in the increased durability of stainless-steel net electrodes compared to textile electrodes, which tend to oxidize more rapidly. Showing time lapse ERT data of e.g., hourly measurements could further underline advantages of the stainless-steel net electrodes and enhance the relevance of the article. Clearly, this would require considerable additional effort. Nonetheless, without any further developments compared to Bast et al. (2024) the study lacks substantial new data or insights that would justify its publication as an independent contribution to the field. Additionally, I would include a comparative analysis between net and textile electrodes to clearly demonstrate the advantages of the new design over both textile and traditional spike electrodes. I also suggest revising the Discussion section; vague terms such as "good contact resistances" should be replaced and the discussion points need to be better supported with relevant literature.

Taking these concerns along with a number of specific comments and technical corrections listed below into account, I recommend accepting this manuscript after major revisions.

**Specific comments and technical corrections**

Line 10: I would also include talus slopes, debris-covered bedrock, moraines, debris-covered glaciers in the description, which also consist of blocky to coarse blocky surfaces.

Line 12: we have successfully tested alternative electrodes that are more robust, lighter (and cheaper than the recently proposed conductive textile electrodes

Abstract in general: How do you prove that these suggested electrodes are better than others, because the reciprocal error is lower for these electrodes? You also need to address your main findings and results within the abstract, not only in the conclusion.

Lines 16-19: different types of study environments: in which environments is ERT a widely applied method and why? "as data acquisition is usually relatively rapid and not particularly complicated" is not the first reason why ERT is used in all these environments. I would rather start with a sentence as "ERT is a non-destructive geophysical method that provides continuous and detailed 2D imaging of variations in the electrical properties of the subsurface.", shortly describe the method and list a few applications.

Line 22: I would cite here more fundamental studies concerning galvanic coupling between electrodes and ground.

Lines 22-24: I suggest going into more detail concerning data quality, high contact resistances, what is "high" and what is the influence of high contact resistances on ERT data?

Line 29: What are excellent mechanical properties? High mechanical strength and durability?

Lines 30-32: I would change it to: Accordingly, for several decades, electrodes in ERT surveys have commonly been produced as (or made out of) round stainless-steel spikes …

Line 36: change "is guaranteed" to "was guaranteed", and "were added"

Line 39: I would change "finally easily removed" to "easily removed after the measurements"

Lines 39-40: I would go into more detail here, what are the main aspects and conclusions of this study, are these textile electrodes reliable compared to traditional electrodes?

Line 55 and line 61: I would delete "a chaotic electrode arrangement/mixture of"

Line 66: Why did you use these dimensions? Did you test different dimensions of the square and the size of the sponges and analysed contact resistances/ injected current? You do not show the connectors to the cables. How do you connect the steel net electrodes to the cables? Do you have problems with oxidation of the connectors?

Line 73: … with different skips as described in Pavoni et al. (2023), …

Line 74: … and included reciprocal (interchanged current and potential dipoles) measurements… I would somewhere include a short description of normal and reciprocal measurements. How did you compute the reciprocal error? Why did you take the reciprocal error as an estimate to evaluate data uncertainty? Explain in more detail.

Line 77: The two different electrode types were placed at the same location between blocks and boulders.

Line 89: How did you define the data error within the inversion? I suggest describing it here.

Line 94: 100 kΩ is still not "optimal", I would delete/change this word.

Lines 99-102: I would somewhere (in the Appendix, in a table) add additional information on the three sites. What is the active layer depth or thaw layer depth of the measurement day for the two rock glaciers? You write that the ERT results reconstruct the known structure of the landslide deposit (Weidinger et al., 2014) and confirm the presence of a frozen layer at the rock glacier (Bast et al., 2024), I would then summarize these validation data for a better comparison to the results.

Line 111: Change "excellent" to "high"

Line 112: "In the first case $R^2$ = 0.91, while for the inverted resistivities $R^2$ = 0.93." This sentence has no verb, please summarize with previous sentence.

Line 116: …. more than half of the electrodes yield contact resistances > 200 kΩ.

Line 118: inversion process

Line 119: delete "measured"

Results section in general: I suggest adding additional pseudosections to visualize the spatial consistency in the apparent resistivity readings as well as the position of the removed quadrupoles for the different electrode types. Additionally, it would be interesting to show injected current for the different electrode types.

Line 126: I suggest revising this sentence and providing a more detailed discussion on the effect of high contact resistances on ERT data quality, supported by references.

Line 128: What is good? I would use low or high

Line 137-139: You have the reciprocal error as an estimate of data error to evaluate data quality/uncertainty. Why don't you refer to this here?

Line 137: "better" → "lower"

Line 138: "greater" → "higher"

Line 139: "best" → "lowest"

Lines 142-143: delete "although excellent"

Line 145: poorer → lower

Lines 146-148: I suggest revising these lines to provide a clearer comparison of your findings with existing studies. Please clarify which high-resistivity structure you are referring to.

Lines 149-151: How can it be demonstrated that the stainless-steel net electrodes yield results equivalent to those of conductive textile electrodes, given that no direct comparison is provided within your study? Please explain in more detail.

Line 156: induction polarisation measurements → induced polarization measurements!

Figures: The font size in some parts of the figures is too small and should be increased to ensure readability.

I would use abbreviations for apparent resistivity, electrical resistivity, contact resistance.

---

## Author Comment (AC3)

[Figure]

**Figure 2.** Histograms (a), (d), and (g) compare grounding resistances [kΩ] recorded at the Marocche di Drò, Sadole, and Flüela test sites, respectively, using traditional stainless-steel spike electrodes with sponges (blue) and the proposed stainless-steel net electrodes with sponge inserts (red). Panels (b), (e), and (h) show the corresponding injected electric currents [mA], while (c), (f), and (i) present the reciprocal error [%] of the quadrupoles for the same sites and electrode types. All electrodes were moistened with the same amount of saltwater and placed at comparable positions between surface boulders (see Fig. 1b–c). Panels (j), (k), and (l) illustrate the contact resistances (first 24 electrodes, as in panel a), injected currents, and reciprocal errors for datasets acquired at the Sadole site in June 2024 (orange) and June 2025 (yellow) along the permanent ERT monitoring line using the stainless-steel net electrodes.

---

## Author Response (AR1)

**EGUSPHERE-2025-405**

**Brief communication**

*by Pavoni et al. 2025*

*Paper added to special issue: Emerging geophysical methods for permafrost investigations: recent advances in permafrost detecting, characterizing, and monitoring*

**Point by Point reply to Editor**

*Dear Editor,*

*We would like to thank the 2 Reviewers for their valuable comments. We have modified our manuscript accordingly. A detailed response to the reviewers' comments is provided in the discussion. Here is a point by point reply to your comments. For clarity, our replies are in italics.*

Dear authors,
Both reviewers consider your manuscript useful and suitable for the special issue, although they note its limited degree of innovation and suggest major revision by adding additional data (induced polarisation and/or monitoring) and the corresponding discussion.
While I agree with the reviewers that the demonstration of a new electrode type does not necessarily warrant publication, I find the statistical data interesting and useful. In particular, your suggested additions of pseudosections and the repetitive measurements should raise the level of relevance sufficiently to justify publication.

*Dear Professor Hördt,*
*We would like to thank you for your valuable comments and suggestions. We are grateful for your understanding of our aims and recognition of their relevance, particularly within the framework of the special issue.*
*We are firmly convinced that steel net electrodes can significantly improve and facilitate ERT investigations on rock glaciers, as recognized by other colleagues—who regularly work in these challenging environments—at the conferences where we recently discussed our work. As noted to the reviewers, steel net electrodes overcome the disadvantages of textile ones—especially the high material costs, oxidation causing performance loss, weight comparable to steel rods, and poor fabric resistance, which can be cut when the bags are wedged between blocks. Even if far from thrilling advances, our simple message is that net electrodes are more cost-effective, resist oxidation, and—as suggested by you and now shown in the revised manuscript—maintain excellent long-term performance for monitoring. Thus, steel net electrodes combine the robustness and longevity of steel rods with the advantages of textile ones, while adding the two core benefits of lower cost and weight.*

I have a few own comments which I also like you to consider in the revised version:
1) The textile electrodes have been patented in Germany. The number is DE 10 2021 110 721, under which the patent can be found. I admit that it is not easy to know about, but it

exists and also should be mentioned and referred to. While the fact of the patent itself may not be so relevant, there is one difference to the existing electrodes: the patented textile electrodes are refillable. The main idea is that only the refillable textile bags need to be carried, and any material that is found in place (sand, water, mud) may be used as filling material. This possibility has not been exploited or appreciated in the previous publications, but I believe it is relevant in this context.

While I see that the necessity to refill is a disadvantage, the "bags only" are much lighter than the filled bags, and also the volume would be much smaller than the net+sponge solution. Of course, the cost + corrosion disadvantages remain. in any case, the discussion should not ignore this option.

*We will include the patent number of the textile electrode in the revised manuscript as you suggested. As you stated, this was not an information we had access, thanks for sharing it. The proposal to "construct" the textile electrodes in situ to reduce weight and volume during transport is interesting. However, in our opinion, it has strong practical limitations for several reasons. Fine material (sand, silt or mud) is rarely available on blocky rock glaciers or other landforms in high mountain environments. Even if assembly is possible, the electrodes would regain their original weight and the problem would reoccur if multiple lines needed to be deployed on the rock glacier in the same survey, as is usually the case. The most problematic issue, however, is the time loss: we would first have to search for fine material to fill the bags, then assemble 48 (or even 72–96) electrodes, and finally disassemble them at the end of the measurement day. Filling the bags with water may be an optimal solution but, unfortunately, on rock glaciers, water is very precious for wetting the electrodes before measurements to improve contact resistances and transport from front spring would means having to climb up and down the rock glacier multiple times. Anyway, the same in situ assembly idea could be applied to the steel net electrode too, and deserves to be tested as you suggested.*

2) In your response to the reviewers, the argument that this is a brief communication is mentioned several times to explain why additional information cannot be provided. While I agree that a brief communication is an appropriate format for this material, it should not serve as a justification for omitting appropriate responses to the reviewers' suggestions. I do see some potential for shortening in other sections; my feeling is that chapters 1, 3 and 4 could be written more concisely.

*In the new version of the manuscripts there are some shortening and we try to address all the comments of the Reviewers. None of the suggestions was omitted. We have added additional results to Figures 2 and 3, we were required to expand the text, which in turn made it necessary to shorten some sections related to the methodology and site characterization (already presented in the cited references). We felt the need to repeatedly clarify to Reviewer 1 that the manuscript under review is a ''Brief Communication'', as she/he did not seem to grasp our limitation, or may not have checked the specific guidelines for this format. Include all the details requested by the reviewer cannot in fact fit the four-page limit. We focused on the key elements necessary to ensure the clarity of the work (as Reviewer 2 correctly understood), while providing clear and relevant references for readers to explore more detailed information (mostly regarding the ERT method and site characterization). We hope this new version meets both the response needs and the editorial rules.*

3) I recommend to carefully re-read the reviewers' suggestions, it seems that some of them have been misunderstood. For example, the comment "Line 89: How did you define the data error within the inversion?" was answered by "...we think that in the chapter we clearly explained the choice of the reciprocal error threshold: At each test site, the datasets were filtered using a  reciprocal error threshold that allowed...". The point was missed here, because the question was on the data errors entering the inversion, and not on the filtering procedure.

*We are sorry for this misunderstanding, the revised version took in consideration this comment. In Section 3.3, we provided a general explanation of how we selected the reciprocal error threshold for data filtering. Then, in Section 4, we report the specific inversion error for each site, derived from the analysis of the reciprocal error, for each dataset. Hence, we provided the specific data errors in the inversion to ensure repeatability. We revised this part accordingly.*

4) Make sure that your replies should normally also lead to a change in the final version of the manuscript. For example, your reply "We have already published a study in which we explored the effect of performing ERT measurements in debris-block surface environments, both with and without wetting the electrodes with saltwater (Pavoni et al., 2022)", should be followed by a corresponding revision. Note that if you find that a reviewer misunderstood your intentions, this could at least partly be due to imprecise formulation.

*We agree and always consider Reviewer suggestions. On the other hand, while every comment needs a response, not every suggestion from one reviewer must necessarily lead to changes in the revised version of the manuscript, since it may not meet other reviewers' comment. In this specific case, our intention was simply clarifying that some of the topics suggested had already been addressed in the cited reference. However, as stated in our response, we truly appreciated Reviewer 2 suggestion to present results demonstrating the durability of the steel net electrodes. Accordingly, we included two new panels in Figure 2, where we compare the results from the Sadole rock glacier site taken one year apart. We believe this suggestion brought to a very relevant improvement of the message, as you suggested too.*

Looking forward to the revised manuscript.
Sincerely,
Andreas Hördt
editor

*Once again, we thank both Reviewers for their constructive comments and questions. We are convinced that the revised version of our manuscript, improved and strengthened in structure and presentation by the relevant suggestions of the Reviewers and Editor, can be further considered for publication in the special issue.*

**Point by Point reply to Reviwer #1**

*We would like to thank the anonymous reviewer for his/her comments. While we carefully considered his/her suggestions, we disagree with the assertion that there is a lack of novelty compared to previous papers. Our work Bast et al. (2024) was, in fact, substantially different: to verify the reliability of the conductive textile electrodes proposed by Buckel et al. (2023). This work, which is limited in scope as it is a brief communication, proposes a new approach to electrodes that provides an improved solution for performing ERT surveys in coarse, blocky environments.*

**Summary**

The manuscript presents a study on a newly developed electrode design, i.e., stainless steel-net electrodes, which the authors propose for electrical resistivity tomography (ERT) measurements in coarse blocky environments. The electrodes facilitate and accelerate ERT surveys and are cheaper and lighter compared to conventional stainless-steel spike electrodes. The authors demonstrate that contact resistances and reciprocal errors are lower when using stainless-steel net electrodes compared to conventional stainless-steel spike electrodes. Correlations of apparent resistivity values between the two electrode types are high for an exemplary landslide deposit and an Italian rock glacier (with an R2 between 0.91 and 0.99) and slightly lower for a Swiss rock glacier (R2=0.8). The inversion results are similar for both electrode types and successfully reconstruct the known internal structure of the landforms. The results are clearly presented and highlight the relevance of lightweight, easily deployable equipment in harsh alpine terrain with limited accessibility.

However, upon closer examination, the manuscript offers limited novelty and primarily reiterates concepts and methodologies previously presented, particularly in the study by Bast et al. (2024). One of the main concerns lies in the similarity between the figures of the two articles. The representations and analysis of the data are almost identical, with no substantial additions, enhancements or new interpretations. For instance, the authors could have included additional pseudosections to visualize the spatial consistency in the apparent resistivity readings as well as the position of the removed quadrupoles for the different electrode types. In my opinion, the results related to the application of stainless-steel net and textile electrodes could have been presented together in one publication, as the study sites are identical and the data analysis and structure is very similar.

To strengthen the manuscript, I recommend that the authors explore the usability of such stainless-steel net electrodes for induced polarization measurements, as suggested in the Discussion section. Such an investigation would offer a clear advancement over previous research. Another benefit highlighted by the authors consists in the increased durability of stainless-steel net electrodes compared to textile electrodes, which tend to oxidize more rapidly. Showing time lapse ERT data of e.g., hourly measurements could further underline advantages of the stainless-steel net electrodes and enhance the relevance of the article. Clearly, this would require considerable additional effort. Nonetheless, without any further developments compared to Bast et al. (2024) the study lacks substantial new data or insights that would justify its publication as an independent contribution to the field. Additionally, I would include a comparative analysis between net and textile electrodes to clearly demonstrate the advantages of the new design over both textile and traditional spike electrodes. I also suggest revising the Discussion section; vague terms such as "good contact resistances" should be replaced and the discussion points need to be better supported with relevant literature.

Taking these concerns along with a number of specific comments and technical corrections listed below into account, I recommend accepting this manuscript after major revisions.

*AUTHORS REPLY: while we agree that the presented study shows a similar analysis to that in our previous work (Bast et al., 2024), this brief communication has a completely different aim. Rather than proposing a comparative analysis of different electrode approaches, this work aims to propose a*

*new type of electrode with significant improvements compared to the recently tested conductive textile electrodes (Buckel et al., 2023). As highlighted in Bast et al. (2024), conductive textile electrodes are a reliable tool for facilitating the acquisition of ERT measurements on coarse-blocky surfaces such as rock glaciers and landslide deposits. Textile electrodes can be easily inserted and removed between the blocks without the need for steel spikes to be hammered in, which significantly speeds up the preparation of ERT arrays. Nevertheless, in our discussion in Bast et al. (2024), we identified several important problems related to the application of these conductive textile electrodes that this work intends to overcome: i) The same weight of steel spike electrodes (50 steel spike electrodes or 50 conductive textile electrodes weigh about 15 kg in total); ii) The oxidation problem related to the conductive metals used to make the textile (copper and nickel), which will drastically reduce the performance of the electrodes after few surveys; iii) The cost of making (or replacing damaged ones) the textile electrodes (15 euros each, mainly due to the cost of the conductive textile, i.e. 750 euros for a set of 50 electrodes);iv) the conductive textile is fragile and prone to be cut by rough surfaces.*

*As we highlighted in the submitted BRIEF communication, the proposed steel-net electrodes clearly overcome all these problems: i) each electrode is just 50 g (50 electrodes, which can easily fit in a traditional medium size mountain bag as the textile electrodes, are about 2.5 kg), ii) the net is realized with stainless steel, therefore we don't face any oxidation problem, and this means that the high performance of the electrodes is guaranteed in future surveys; iii) producing the steel-net electrodes is also much more advantageous from an economic point of view, the net can be easily found in hardware stores, and the cost to produce an electrode is around 2 euros, about 100 euro for 50 electrodes; iv) finally, the mechanical resistance of the electrodes is improved, these new steel-net electrodes do not break and last a long time.*

*Taking all this into account, we believe that the proposed electrodes offer some significant advantages: i) The low weight of the electrodes is a valuable improvement for researchers working in high mountain environments who are accustomed to carrying all the ERT equipment without the support of vehicles (e.g. helicopters); ii) once the electrodes have been made, there is no need to replace them due to oxidation (or to spend time drying the textile at the end of each survey to reduce oxidation) or breakage; iii) even if they need to be replaced, the cost of the net electrode is low and the material is easily available. Therefore, our new solution is a significant improvement on textile electrodes: we have all the advantages presented by Buckel et al. (2023), while overcoming the issues identified by Bast et al. (2024).*

*While we agree that the analysis is similar to that of Bast et al. (2024), also here we have considered and compared the relevant parameters in an ERT survey, namely contact resistances, the quality of the measured datasets (via reciprocal error) and the results (i.e. the inverted resistivity model). We could add the pseudo-sections, but we have already plotted the linear regression of the measured apparent resistivities to demonstrate the consistency between the datasets obtained using different electrodes. However, Cs Brief Communications are limited to 3 figures, so we must make a precise and concise selection of what to show readers. In our opinion, the structure of the figures is not exactly the same as that shown by Bast et al. (2024). While it is true that we present and compare the same parameters, this is unavoidable for an ERT survey performance evaluation. Regarding the choice of test sites, the work is not finalised to characterise the subsurface structure/composition, which is already well known, but rather to verify the reliability of the proposed electrodes for acquiring ERT surveys in these environments.*

*This analysis could not be included in Bast et al.'s (2024) previous work since the steel-net electrodes were developed after the manuscript had been submitted. As the reviewer correctly stated, the performance of textile and stainless-steel net electrodes is substantially the same, as verified at the same study sites. Therefore, we preferred to compare our new electrodes with the commonly used steel spikes coupled with sponges (soaked in salt water), the traditional approach to acquiring ERT*

*datasets in rock glacier environments, and consequently a more representative way to confirm the reliability of the new electrodes.*

*In each test-site we performed both ERT and time-domain IP measurements. However, the IP data error (considering reciprocal errors and fitting of the curve relaxation) was too large with both the electrode types, and hindered a relevant and significant IP analysis, in our opinion.*

*The Syscal acquisition parameters were configured for the IP to the best of our knowledge, with a 2-second injection-measurement period and 20 custom sampling intervals increasing from 20 ms to 200 ms. A specific dipole-dipole sequence was used to avoid the use of polarized electrodes as potential electrodes. A 10-minute break between direct and reciprocal measurements was allowed to provide sufficient time for the depolarization of the electrodes. Hence, our understanding is that the very high IP errors were related to the arrangement of the ERT cables. In fact, multicore cables were used without separating current and potential arrays, as extensively discussed by Maierhofer et al., (2022). The IP errors we estimated are in line with this study (see for example the number of discharged measurements in their figure 2), confirming that IP surveys in such rocky and challenging environments likely require separating current and potential cables. Note that, in our study, the choice of not separating the cables reflected the focus on ERT data quality and kept our tests comparable with the vast majority of similar studies and applications.*

*As an example, from the Marocche test site, the following figure summarizes the IP error analysis. The figure shows how both spike and net electrodes have extremely high reciprocal errors for the chargeability (y axis), also relative to the resistance reciprocal errors (x axis). The scattering highlights how resistance and IP errors are not correlated. The colours represent the error estimated from the fitting of the IP decay curve analysis, following the algorithm suggested by Orozco et al., (2018), which is also not clearly related to the reciprocal errors in this case. We also tried to ignore early and late decay curve samples, but this did not have significant effects.*

[Figure]

*In conclusion, our understanding is that these IP errors are 1) too large to support a relevant IP analysis, 2) very complex, with no clear relationships between the resistance and IP reciprocal errors, nor between decay curve analysis and reciprocal errors, nor between spike and net electrodes; and 3) in line with previous studies using non-separated cables in such environments, which also limits the possible novelty.*

*Therefore, as the Reviewer can clearly understand, we decided to not insert the comparison between the collected IP measurements, but we will run new tests in future using different cables for injecting electrodes and potential electrodes, to improve the quality of the IP measurements.*

*Regarding the oxidation problem of the textile electrodes, we agree that it could be measured using a time-lapse configuration, but only with very long-time monitoring. We doubt in fact that we could*

*verify a performance drop using hourly measurements, given that we have used them with excellent results in several campaigns during the 2023 summer season. The degradation process is much slower, becoming apparent after several weeks of continuous use of the textile electrodes. Anyway, the aim of this work was not to verify the 'lifetime' of textile electrodes, but rather to propose a new type of electrode that does not oxidise.*

*Taking all this into account, while we respect the opinion of the anonymous Reviewer but we believe that this BRIEF communication explores significant advantages for permafrost ERT community, since the proposed steel-net electrodes represent a clear cheap and effective improvement.*

**Reviewer Specific comments and technical corrections**

- Line 10: I would also include talus slopes, debris-covered bedrock, moraines, debris-covered glaciers in the description, which also consist of blocky to coarse blocky surfaces.

*Reply: We agree with the suggestion of the reviewer, and we will modify it in the revised manuscript.*

- Line 12: we have successfully tested alternative electrodes that are more robust, lighter (and cheaper than the recently proposed conductive textile electrodes

*Reply: We agree with the suggestion of the reviewer, and we will modify it in the revised manuscript.*

- Abstract in general: How do you prove that these suggested electrodes are better than others, because the reciprocal error is lower for these electrodes? You also need to address your main findings and results within the abstract, not only in the conclusion.

*Reply: We would like to highlight to the reviewer that, in the abstract or elsewhere in the manuscript, we never presented the proposed steel-net electrodes as superior to other types, such as steel spikes or textile electrodes. In the abstract, which is limited to 100 characters, we just briefly presented the net electrodes and their advantages compared to textile electrodes. The reliability of the latter has already been verified in Bast et al. (2024). Furthermore, in both the abstract and the introduction, we highlighted that we successfully tested the performance of the steel-net electrodes. We believe the details of the experiment can be found in the main body of the manuscript, as well as in the discussion and conclusions.*

- Line 22: I would cite here more fundamental studies concerning galvanic coupling between electrodes and ground.

*Reply: The work by Pavoni et al. (2022) clearly highlights the link between galvanic contact (or contact resistance) and data quality; in fact, the work focuses entirely on this. We refer to this work and the references cited within it for further specifications, since including a long list of redundant references is beyond the scope of a Brief Communication.*

- Lines 22-24: I suggest going into more detail concerning data quality, high contact resistances, what is "high" and what is the influence of high contact resistances on ERT data?

*Reply: as a BRIEF communication we have with a limited number of pages (4), references, and figures, that limit to go into detail of so common aspects in ERT techniques. Further details about the ERT method, data quality, contact resistance relations etc. con be found in more specific 'fundamental' manuscripts that are insert in the references list (e.g., Binley (2015), Tso et al. (2017), and Binley and Slatter (2020)).*

- Line 29: What are excellent mechanical properties? High mechanical strength and durability?

*Reply: We agree with the suggestion of the Reviewer and we will modify it in the revised manuscript.*

- Lines 30-32: I would change it to: Accordingly, for several decades, electrodes in ERT surveys have commonly been produced as (or made out of) round stainless-steel spikes …

*Reply: We agree with the suggestion of the Reviewer and we will modify it in the revised manuscript.*

- Line 36: change "is guaranteed" to "was guaranteed", and "were added"

*Reply: We agree with the suggestion of the Reviewer and we will modify it in the revised manuscript.*

- Line 39: I would change "finally easily removed" to "easily removed after the measurements"

*Reply: We agree with the suggestion of the Reviewer and we will modify it in the revised manuscript.*

- Lines 39-40: I would go into more detail here, what are the main aspects and conclusions of this study, are these textile electrodes reliable compared to traditional electrodes?

*Reply: We thank the Reviewer for this comment, and we will strengthen our concept. We successfully tested the performance of the steel-net electrodes against the traditional ones as clearly visible in our figures. We will try to rephrase the details of the study in the main body of the manuscript, considering again that we submitted a Brief Communication, and consequently we have limited space that does not admit repetitions.*

- Line 55 and line 61: I would delete "a chaotic electrode arrangement/mixture of"

*Reply: We thank the reviewer for his/her suggestion, but the surface of our landslide deposits and of the rock glaciers are effectively composed of a chaotic mixture of debris and boulders with different sizes.*

- Line 66: Why did you use these dimensions? Did you test different dimensions of the square and the size of the sponges and analysed contact resistances/ injected current? You do not show the connectors to the cables. How do you connect the steel net electrodes to the cables? Do you have problems with oxidation of the connectors?

*Reply: We will add these details. We used a traditional car-wash sponge to create a bag that has the same size of the textile-electrode proposed by Buckel et al. (2023). Since the collected datasets shows acceptable contact resistance values and reliable data quality (based on the reciprocal error), we didn't consider using different size of the bag. As shown in figure 1b and 1c, the steel-net electrodes are connected to the cable with traditional crocodiles and we don't have any oxidation problem with the connectors.*

- Line 73: … with different skips as described in Pavoni et al. (2023), …

*Reply: We agree with the suggestion of the Reviewer and we will modify it in the revised manuscript.*

- and included reciprocal (interchanged current and potential dipoles) measurements… I would somewhere include a short description of normal and reciprocal measurements. How did you compute the reciprocal error? Why did you take the reciprocal error as an estimate to evaluate data uncertainty? Explain in more detail.

*Reply: as previously, we would like to highlight that this work is submitted as a BRIEF communication with a limited number of pages (4), references, and figures. Further details about the ERT method, data quality, and contact resistances con be found in more specific ERT 'fundamental' manuscript that are insert in the reference list. In the data processing chapter (3.3) we also explained the choice of the reciprocal error threshold.*

- Line 77: The two different electrode types were placed at the same location between blocks and boulders.

*Reply: We agree with the suggestion of the Reviewer and we will modify it in the revised manuscript.*

- Line 89: How did you define the data error within the inversion? I suggest describing it here.

*Reply: We thank the reviewer for his suggestion, but we think that in the chapter we clearly explained the choice of the reciprocal error threshold: "At each test site, the datasets were filtered using a reciprocal error threshold that allowed to obtain a section of apparent resistivities with a homogeneous distribution of measurement points…"*

- Line 94: 100 kΩ is still not "optimal", I would delete/change this word.

*Reply: We respect the opinion of the reviewer, but, in our experience and in ERT permafrost literature, reaching 100 kΩ of contact resistance value with large boulders at the surface can be considered a very good result that allows to acquire reliable ERT datasets in this kind of environments. In the revised manuscript we can change 'optimal' with 'acceptable'.*

- Lines 99-102: I would somewhere (in the Appendix, in a table) add additional information on the three sites. What is the active layer depth or thaw layer depth of the measurement day for the two rock glaciers? You write that the ERT results reconstruct the known structure of the landslide deposit (Weidinger et al., 2014) and confirm the presence of a frozen layer at the rock glacier (Bast et al., 2024), I would then summarize these validation data for a better comparison to the results.

*Reply: We agree with the Reviewer, but as said, editorial guidelines for BRIEF Communication give a limited number of pages (4) and no appendixes are allowed. The information requested by the reviewer can be easily found in other works that we cited in the manuscript. The target of this work was in fact not to characterize the well-known structure of these sites.*

- Line 111: Change "excellent" to "high"

*Reply: We agree with the suggestion of the Reviewer and we will modify it in the revised manuscript.*

- Line 112: "In the first case R2 = 0.91, while for the inverted resistivities R2 = 0.93." This sentence has no verb, please summarize with previous sentence.

*Reply: We agree with the suggestion of the Reviewer and we will modify it in the revised manuscript.*

- Line 116: …. more than half of the electrodes yield contact resistances > 200 kΩ.

*Reply: We agree with the suggestion of the Reviewer and we will modify it in the revised manuscript.*

- Line 118: inversion process

*Reply: We agree with the suggestion of the Reviewer and we will modify it in the revised manuscript.*

- Line 119: delete "measured"

*Reply: We agree with the suggestion of the Reviewer and we will modify it in the revised manuscript.*

- Results section in general: I suggest adding additional pseudosections to visualize the spatial consistency in the apparent resistivity readings as well as the position of the removed quadrupoles for the different electrode types. Additionally, it would be interesting to show injected current for the different electrode types.

*Reply: We thank the reviewer for his/her suggestion and, for each site, we will add in Figure 3 a plot of the pseudosection with the difference between the resistance (R[Ω]) values measured with the two types of electrodes. Note that, at each site, only the quadrupoles common to the two datasets after filtering were used for the inversions, as in Bast et al. (2024). We will highlight this better in the revised manuscript. In the results chapter we focused on describing the results and highlighting the high correlation between the data measured with the different types of electrodes. Furthermore, in Figure 2 we added the comparison between the measured injected current values.*

[Figure]

*Figure 3: a) Scatterplot with corresponding regression lines (red dotted lines) and R²-values of the apparent resistivity values (ρa) measured at Marocche di Drò test site with the traditional spike electrodes composed of stainless-steel (coupled with sponges) and the newly proposed stainless-steel net electrodes. d) pseudo-section plotting the ratio between resistance values measured with the spike electrodes and steel-net electrodes (considering the common quadrupoles in the filtered datasets used for the inversion process).g) Scatterplot with corresponding regression lines (red dotted lines) and R²-values of the inverted resistivity values (ρ) obtained from the datasets measured at Marocche di Drò test site with the traditional stainless steel-spike electrodes (coupled with sponges) and the proposed stainless steel-net electrodes. j) Inverted resistivity model obtained from the datasets measured with the traditional stainless steel-spike electrodes at the Marocche di Drò test site. m) Inverted resistivity model obtained from the datasets measured with the proposed stainless steel-net electrodes at the Marocche di Drò test site. p) Color bar scale for the inverted resistivity models of the Marocche di Drò test site. b), e), h), k), n) and q) are as a), d), g), j), m) and p) but for the Sadole rock glacier test site. c), f), i), l), o) and r) are as a), d), g), j), m) and p) but for the Flüela rock glacier test site.*

[Figure]

*Figure 2: The histograms (a), (d), and (g) illustrate the comparison of contact resistances [kΩ] recorded respectively at the Marocche di Drò, Sadole, and Flüela test sites with the traditional spike electrodes made of stainless steel and combined with sponges (blue bins) and the newly proposed stainless-steel net electrodes that include a sponge (red bins). The histograms (b), (e) and (h) display the comparison of the reciprocal error [%] of the quadrupoles measured respectively at the Marocche di Drò, Sadole, and Flüela test sites with the traditional spike electrodes made of stainless steel and combined with sponges (blue bins) and the newly proposed stainless-steel net electrodes that include a sponge (red bins). The histograms (c), (f) and (i) display the comparison of the measured injected electrical current [%] of the quadrupoles measured respectively at the Marocche di Drò, Sadole, and Flüela test sites with the traditional spike electrodes made of stainless steel and combined with sponges (blue bins) and the newly proposed stainless-steel net electrodes that include a sponge (red bins). Both types of electrodes were wetted with the same amount of salt water and approximately placed at the same locations between the blocks and boulders (Fig. 1b and 1c).*

- Line 126: I suggest revising this sentence and providing a more detailed discussion on the effect of high contact resistances on ERT data quality, supported by references.

*Reply: again we are forced to a limited number of pages (4) which does not admit too details. The information requested by the reviewer can be easily found in other works, and, as correctly suggested, we will add some references in the revised manuscript.*

- Line 128: What is good? I would use low or high

*Reply: We agree with the suggestion of the Reviewer and we will modify it in the revised manuscript.*

- Line 137-139: You have the reciprocal error as an estimate of data error to evaluate data quality/uncertainty. Why don't you refer to this here?

*Reply: We thank the Reviewer for the suggestion, we will add this aspect.*

Line 137: "better" → "lower"

*Reply: We agree with the suggestion of the Reviewer and we will modify it in the revised manuscript.*

- Line 138: "greater" → "higher"

*Reply: We agree with the suggestion of the Reviewer and we will modify it in the revised manuscript.*

- Line 139: "best" → "lowest"

*Reply: We agree with the suggestion of the Reviewer and we will modify it in the revised manuscript.*

- Lines 142-143: delete "although excellent"

*Reply: We agree with the suggestion of the Reviewer and we will modify it in the revised manuscript.*

- Line 145: poorer → lower

*Reply: We agree with the suggestion of the Reviewer and we will modify it in the revised manuscript.*

- Lines 146-148: I suggest revising these lines to provide a clearer comparison of your findings with existing studies. Please clarify which high-resistivity structure you are referring to.

*Reply: We thank the reviewer for his/her suggestion, we can improve this sentence to clarify that the high-resistivity structure found in the resistivity models is interpreted as the frozen-permafrost layer.*

- Lines 149-151: How can it be demonstrated that the stainless-steel net electrodes yield results equivalent to those of conductive textile electrodes, given that no direct comparison is provided within your study? Please explain in more detail.

*Reply: We agree, we can remove that sentence since we don't have a direct comparison. Anyway, as discussed in the site description chapter (2), in this work we performed the test-transect on the same survey lines that were investigated by Bast et. al (2024), and the results are perfectly in agreement.*

- Line 156: induction polarisation measurements → induced polarization measurements!

*Reply: We agree with the suggestion of the Reviewer and we will modify it in the revised manuscript!*

- Figures: The font size in some parts of the figures is too small and should be increased to ensure readability.

*Reply: We agree with the suggestion of the Reviewer and we will modify it in the revised manuscript.*

**Point by Point reply to Reviwer #2**

We thank Reviewer #2 for reviewing our manuscript and providing valuable suggestions to improve the quality of our manuscript. The reviewer clearly understood our aims and intentions, and we are aligned with the critical comments raised by the reviewer.

- Reviewer #2: The manuscript describes and evaluates the performance of a modified electrode design that aims to optimize the process of acquiring resistivity measurements in blocky, rocky terrain. The manuscript, submitted in the form of a brief communication, focuses on i) describing the construction and practical advantages of the modified electrode design, and on ii) confirming the reliability of the proposed electrode design in comparison to more established electrode types (stainless steel spikes).

*Authors' reply: The reviewer has clearly grasped the aim of the submitted Brief Communication, which is to propose novel electrodes that facilitate and optimize ERT measurements in high-mountain environments with debris-block surfaces, compared to traditional steel rods and textile bags (Buckel et al., 2023).*

- Reviewer #2: The principle of the proposed electrode design - sachets made of a conductive material filled with a porous matter that holds moisture and allows the sachet to mould to its surroundings - was, as the authors state, inspired by a cited study (Buckel et al. (2023)). The modifications in the present manuscript propose replacing the conductive textile with more robust and cheaper stainless-steel nets, and the fill of fine sand with lighter carwash sponges. These modifications resulted in three main improvements: the proposed stainless steel mesh electrodes are reported to be cheaper, lighter and more durable than the conductive textile electrodes proposed by Buckel et al. (2023) - all being important considerations when preparing for a resistivity survey.

*Authors' reply: Our proposed steel net electrodes provide all the advantages of the textile electrodes introduced by Buckel et al. (2023), whilst addressing the limitations highlighted in the comparative tests presented by Bast et al. (2024), namely the high costs of the conductive fabric, the issue of oxidation affecting the copper-nickel textile, and the considerable weight of the bags (250–300 g each, comparable to traditional steel rods). The new electrodes are made from inexpensive stainless-steel mesh, which reduces production costs and eliminates oxidation concerns. Furthermore, their weight is significantly lower (approximately 50 g each; 50 electrodes = 2.5 kg), making them much easier to transport, for example, in a medium-sized mountain backpack, with substantially less physical effort. In our opinion, this represents a major advantage for researchers conducting ERT measurements in challenging and remote environments such as rock glaciers.*

Reviewer #2: The manuscript delivers on its two main objectives: in terms of description of the proposed electrode design, it is well described and its practical advantages are clearly stated. In terms of evaluating the performance and reliability of the proposed electrodes, the manuscript reports on a number of well-presented and well-established, and thus easily inter-comparable metrics, including 'contact' resistances, reciprocal errors, and comparison of the apparent and inverted resistivities acquired with the proposed electrodes vs. well-established steel spike electrodes. These tests show that the proposed electrodes yield results equivalent to those achieved with spike electrodes in terms of the quality of the measured resistivity datasets. The proposed electrodes do not appear to significantly and consistently improve the

electrode 'contact' resistances (which was, however, not stated among the goals of the experiment).

*Authors' Reply: As correctly noted, the aim of this work is not to propose new electrodes that improve contact resistance, an inherently challenging parameter in such study environments, but rather to provide a cost-effective and durable alternative to the recently introduced textile electrodes by Buckel et al. (2023). Through our tests and comparison of key parameters in ERT acquisition (contact resistance, injected current, measured apparent resistivity, data quality assessed via reciprocal error, and the inverted resistivity models), we have demonstrated the reliability of net electrodes for surveys in debris-block surface environments such as rock glaciers. Net electrodes combine the advantages of both traditional steel rods and textile electrodes: they allow for quick and easy installation and removal of ERT transects in blocky terrains (as textile electrodes do), while also offering high mechanical strength, resistance to oxidation, and relatively low cost. Moreover, net electrodes present a significant additional advantage over both traditional and textile electrodes: their remarkably low weight. As previously highlighted, each net electrode weighs only 50 g, significantly reducing the physical effort required to transport them in challenging high-mountain environments.*

- Reviewer #2: I commend the authors for pursuing practical improvements to electrode design, especially for ever-challenging mountain environments, as well as for carefully evaluating the performance of the new electrode design prior to basing any interpretations on it. As the focus of the manuscript is practical innovation, I would suggest exploring opportunities to compensate for the somewhat limited novelty (the key design principles are largely inspired by a previously published study) and increase the impact (the electrodes' grounding qualities match though do not significantly outperform the more established electrode types) of the experiment by expanding the types of applications for which the proposed electrodes are validated. In this context, and especially as the brief communication was submitted for a special issue on 'Emerging geophysical methods for permafrost investigations: recent advances in permafrost detecting, characterizing, and monitoring' it would be relevant to quantify the performance of the proposed electrodes in repeated measurements. This could be as simple as measuring the same profile with the proposed electrodes at the same location right after installation (wetted with saline solution, ideal conditions), after drying out (poor measurement conditions), and after re-wetting naturally e.g. by a rain event (good though less-than-ideal conditions as salts may be progressively washed out of the sponges). I reckon a summary of such an experiment would be relevant for the target audience of the special issue, and could be reported in one paragraph.

*Authors' reply: We have already published a study in which we explored the effect of performing ERT measurements in debris-block surface environments, both with and without wetting the electrodes with saltwater (Pavoni et al., 2022). In that work, we clearly showed that conducting measurements without adding saltwater results in extremely high contact resistance values (several hundred kOhm), which clearly prevent the acquisition of reliable ERT datasets (Pavoni et al., 2022; data quality was assessed through reciprocal error). Furthermore, in Bast et al. (2024), we investigated the effect of using freshwater instead of saltwater for wetting the electrodes. In that study, the site where both traditional and textile electrodes were wetted with freshwater showed significantly higher contact resistance values compared to the two test sites where saltwater was used.*

*In accordance with Reviewer #2's suggestion, we believe it would be valuable to include in the submitted Brief Communication some results demonstrating the high mechanical*

resistance and oxidation resistance of the net electrodes, and thus the long-term reliability of their performance.  In June 2024, a permanent ERT monitoring line was installed on the Sadole rock glacier (North Italy) using 48 net electrodes (the first 24 electrodes correspond to those used in the test presented in this study). Over the past year, several datasets have been acquired to investigate the seasonal variations of permafrost in this study area. Figure 2 of the manuscript can be updated to include a comparison of contact resistances, injected electrical currents, and reciprocal error from the datasets acquired in June 2024 and June 2025 (refer to the attached PDF). As shown in panels (j), (k), and (l), after one year, the performance remained essentially unchanged, despite the electrodes having remained in situ. Achieving the same result would not have been possible with textile electrodes (Buckel et al., 2023), as oxidation issues would have inevitably compromised data acquisition. Therefore, we demonstrated that, similar to traditional stainless-steel spikes, net electrodes can also be employed in permanent ERT monitoring lines on rock glaciers.

[Figure]

*Figure 2. Histograms (a), (d), and (g) illustrate the comparison of contact resistances [kΩ] recorded at the Marocche di Drò, Sadole, and Flüela test sites, respectively, using traditional stainless-steel spike electrodes combined with sponges (blue bins) and the newly proposed stainless-steel net electrodes incorporating a sponge (red bins). Histograms (b), (e), and (h) show the comparison of injected electric currents [mA], measured at the Marocche di Drò, Sadole, and Flüela test sites, respectively, with the two electrode types. Histograms (c), (f), and (i) present the comparison of reciprocal error [%] of the quadrupoles measured at the Marocche di Drò, Sadole, and Flüela sites, respectively, using the two electrode configurations. Both electrode types were moistened with the same amount of saltwater and positioned approximately at the same locations between the*

*boulders (see Fig. 1b and 1c). Figures (j), (k), and (l) present a comparison of contact resistances (measured on the first 24 electrodes of the array, corresponding to those shown in Fig. 2a) [kΩ], injected electric currents [mA], and reciprocal error [%] for datasets acquired at the Sadole site in June 2024 and June 2025, along the permanent ERT monitoring line equipped with the newly proposed stainless-steel net electrodes, moistened with saltwater.*

- Reviewer #2: Line 81: What is the protocol for measuring the 'contact' resistances by Syscal-Pro? (type of the electrode test).

*Authors' reply: We agree, and we will provide a more detailed description of the procedure used by the Syscal Pro instrument to measure contact resistance in the modified manuscript.*

- Reviewer #2: I would suggest the authors to consider the advantages of using the term 'grounding resistance' instead of 'contact resistance'. Use of 'grounding' resistance communicates that what's measured during the electrode test is not only the resistance at the contact between the electrode and the embedding medium, but also the effect of geometry of the electrode and properties of the embedding medium, including any alteration zone in the immediate vicinity of the electrode (the saltwater soaked sponges).

*Authors reply: We agree, and we will replace the term contact resistance with grounding resistance in the revised manuscript.*

---

## Referee Report (RR1)

Dear Authors,

Thank you for carefully addressing my comments in the revised version of your manuscript. I appreciate the substantial changes you have made, particularly the inclusion of the additional pseudosections and annual monitoring data, as well as the extended discussion on some important points, which in my opinion significantly enhance the clarity and novelty of the paper. I have no further comments and consider the manuscript suitable for publication in its current form.

Thank you again for your efforts and the careful revision.

---

## Author Response (AR2)

**Dear Professor Hördt,**

We are pleased to hear that the reviewers were satisfied with the revisions we made to the manuscript. In the newly uploaded version, we have included — as you requested — a discussion in the *Discussion and Conclusions* section addressing the possibility of fabricating both types of electrodes (*steel-net* and *textile electrodes*) *in situ*, and we have added the appropriate citation for the textile electrodes.

We hope that the manuscript now meets the criteria for publication.

---

## Author Response (AR3)

Dear Professor Hordt,

Please find attached the final version of our manuscript. We hope it now meets the requirements.